# Model-Driven Analysis of ECG Using Reinforcement Learning

**DOI:** 10.3390/bioengineering10060696

**Published:** 2023-06-07

**Authors:** Christian O’Reilly, Sai Durga Rithvik Oruganti, Deepa Tilwani, Jessica Bradshaw

**Affiliations:** 1Artificial Intelligence Institute of South Carolina, Columbia, SC 29208, USA; orugants@email.sc.edu (S.D.R.O.); dtilwani@mailbox.sc.edu (D.T.); 2Department of Computer Science and Engineering, University of South Carolina, Columbia, SC 29208, USA; 3Carolina Autism and Neurodevelopment Research Center, University of South Carolina, Columbia, SC 29208, USA; jbradshaw@sc.edu; 4Institute for Mind and Brain, University of South Carolina, Columbia, SC 29208, USA; 5Department of Psychology, University of South Carolina, Columbia, SC 29208, USA

**Keywords:** ECG, modeling, reinforcement learning, lognormal, autonomic nervous system, model-driven analysis

## Abstract

Modeling is essential to better understand the generative mechanisms responsible for experimental observations gathered from complex systems. In this work, we are using such an approach to analyze the electrocardiogram (ECG). We present a systematic framework to decompose ECG signals into sums of overlapping lognormal components. We use reinforcement learning to train a deep neural network to estimate the modeling parameters from an ECG recorded in babies from 1 to 24 months of age. We demonstrate this model-driven approach by showing how the extracted parameters vary with age. From the 751,510 PQRST complexes modeled, 82.7% provided a signal-to-noise ratio that was sufficient for further analysis (>5 dB). After correction for multiple tests, 10 of the 24 modeling parameters exhibited statistical significance below the 0.01 threshold, with absolute Kendall rank correlation coefficients in the [0.27, 0.51] range. These results confirm that this model-driven approach can capture sensitive ECG parameters. Due to its physiological interpretability, this approach can provide a window into latent variables which are important for understanding the heart-beating process and its control by the autonomous nervous system.

## 1. Introduction

Heart disease is the leading cause of mortality in the United States and was responsible for 597,689 deaths in 2010, according to the US Centers for Disease Control and Prevention [1]. This number further increased to 693,021 in 2021 [2]. For that reason, tracking how the heart performs is crucial. The electrocardiogram (ECG) is a simple and non-invasive way to record the electrical activity of the heart due to the depolarization and repolarization of its membrane of muscle fibers during the cardiac cycle [3]. It captures the cardiac electrical activity through electrodes placed in direct contact with the skin. The ECG is one of the primary tools to check for irregular heart rhythms (arrhythmia) and help diagnose heart diseases, poor blood flow, and other health issues. Furthermore, ECG activity and its interaction with the respiratory rhythm, a phenomenon known as respiratory sinus arrhythmia, provide a window into the autonomic nervous system (ANS) with potential applications for multiple conditions associated with dysregulated ANS control. In neonatal care, due to the relationship between the immune system and the control of the heart rhythm by the ANS, the ECG is also used to monitor the health of newborns and predict the occurrence of sepsis [4,5].

Over the years, many researchers have developed models to better understand cardiac physiology and its assessment through the ECG, including approaches based on reaction–diffusion, oscillators, and transforms [6,7,8]. The models have specific advantages and disadvantages, so it is important to understand their properties before deploying them in specific applications.

Methods based on reaction–diffusion and oscillators are similar. Although they adopt different perspectives, both involve dynamical models expressed as systems of ordinary differential equations (ODEs). The main drawback of these dynamical models is that their parameters must be limited within a specific range of values to produce realistic ECG signals [6]. These models can exhibit chaotic behavior when their parameters are outside of these ranges. Furthermore, ranges that generate ECG signals with a particular rhythm can be incredibly narrow. The subspace of acceptable parameters can describe a complex manifold in a high-dimensional space (depending on the number of modeling parameters), so simple box constraints are generally insufficient for such applications. Our initial attempts at dynamically estimating parameters for such ODEs to track the modifications of ECG recordings (i.e., inverse modeling) without causing these models to enter a chaotic regime have revealed this problem to be thorny (unpublished results). Thus, although such models can be insightful for forward modeling (i.e., simulating realistic ECG signals), they are difficult to deploy for inverse modeling and inference (i.e., estimating modeling parameter values based on a recorded ECG).

Transform-based modeling has also been proposed to study ECG signals. In this approach, each ECG cycle is decomposed into P, Q, R, S, and T waves, each modeled using elementary functions and represented by their corresponding Fourier series [8,9]. Such an approach may perform well on a stable ECG (i.e., signals displaying no variation from one cycle to the next). However, because it relies on the Fourier series and its decomposition into infinite sinusoidal waves, it is limited for studying temporal (as opposed to spectral) variations in the ECG signal from beat to beat.

Similarly, the individual waves of the PQRST complex can be decomposed and modeled directly in their native space (i.e., as time series). For example, such an approach has been implemented using pairs of normal components [10]. However, a significant body of literature demonstrated the practical and theoretical advantages of using lognormal rather than normal equations for decomposing biosignals. This approach has been pioneered in the context of human movement analysis [11]. In this paper, we build on that literature to develop a novel model of ECG using lognormal decomposition and reinforcement learning. This work paves the way for the analysis of ECGs based on generative, mechanistic models. We aim to use this framework to gain further insights into latent variables involved in normal and abnormal ECG rhythms and better understand how the ANS controls cardiac activity.

In the following sections, we introduce our modeling approach by reviewing (1) the biological process generating ECG signals and (2) the decomposition of biosignals into the sums of lognormals. We then provide further details on our methods, including a description of the participant samples, the ECG preprocessing steps, the PQRST lognormal prototype used to constrain inverse modeling, and the deep reinforcement learning framework used to find optimal lognormal parameters. We then present results for an example use-case, showing how we can use this lognormal decomposition of ECG signal to study the developmental effect of ECG in infants from 1 to 24 months of age. We conclude this paper by discussing the current limitations and future opportunities offered by this novel approach.

## 2. The Generation of ECG

### 2.1. The Cardiac Rhythm and Its Modulation

Understanding the process leading to the generation of biosignals is essential for conducting powerful and meaningful analyses. ECG signals collected by surface leads positioned on the chest are due to the volume conduction of bioelectrical currents in the body. Thus, first, we need to understand how heartbeats lead to current dipoles to properly analyze ECG signals.

The sinoatrial node (SA)—a small region of the upper right atrium made of cells naturally exhibiting cyclic activity—drives the generation of the heart rhythm. The rhythmogenicity of these cells is due to the constant depolarizing effect of leaky sodium channels. However, the time required for these cells to depolarize enough to reach their firing threshold can be increased or decreased by the opening or closing of calcium channels. By regulating the state of these calcium channels through neuromodulation, the autonomic nervous system can control the interval between two beats. This inter-beat interval is also known as the RR interval since it is generally defined as the duration between the peaks of two subsequent R waves (see Figure 1 for the explanation of what R waves are). The balance between the sympathetic and the parasympathetic systems controls the opening/closing of these channels through a molecular cascade involving second messengers. The sympathetic system acts through norepinephrine (neural origin) or epinephrine (distributed through blood circulation). This neuromodulator triggers the phosphorylation of calcium channels, thereby increasing the calcium inflow and causing the earlier depolarization of SA cells (faster cardiac rhythm). The parasympathetic system acts in the opposite way through acetylcholine, which inhibits this process, reducing calcium inflow and prolonging the inter-beat interval. Thus, the central control of the heart and its modulation by various physiological processes (e.g., respiratory sinus arrhythmia) can be captured by modeling the modulation of the calcium current in SA cells.

### 2.2. PQRST Waves

The ECG signal is composed of the repetition of a stereotypical waveform, referred to as the PQRST complex (see Figure 1 for a visual representation). Each heartbeat is responsible for the generation of one such complex. As previously mentioned, cells in the SA node, located on the top-right portion of the right atrium, initiate this process by firing an action potential. This action potential propagates outwardly from one cell to its neighboring atrial myocardial cells through gap junctions, generating a depolarization wave. We can model the far field caused by this wave as coming from an equivalent electric dipole that points toward the lower left side of the heart. This process is responsible for the positive P-wave on a traditional ECG with electrodes placed as shown in Figure 1. This propagation mediated by gap-junction is of moderate speed, causing a relatively low-amplitude, wide-spread positive deflection. A resistive fibrous membrane separates the myocardial cells from the atrial and the ventricular part of the heart, stopping the depolarization wave in the atria from propagating to the ventricles. The atrioventricular (AV) node provides the only window for this depolarization to travel toward the ventricles. The cells of the AV node are specialized for slow conduction, allowing for the delay between atrial and ventricular contraction required to support efficient pumping. Due to the fact that the propagation through the AV node involves a relatively small number of cells, their contribution to electrical currents at distant locations (i.e., at the site of the electrodes) is insufficient to generate an equivalent dipole visible on the ECG. Thus, this phase of the cardiac rhythms results in the flat line characteristic of the PQ segment (also sometimes referred to as the PR segment since the Q component is not always visible on the ECG).

As the depolarization slowly crosses the AV node, it reaches the fast-conducting His–Purkinje network and triggers the depolarization of the lower-left section of the ventricular septum. The resulting wave travels toward the upper part of the septum, causing an equivalent depolarization dipole pointing toward the negative ECG lead and reflected as the fast and low-amplitude negative Q wave. This component is immediately followed by the main depolarization of the ventricles, causing a short-lived and large-amplitude equivalent dipole pointing toward the positive lead and visible on the ECG as the R wave. This wave then propagates to the basal region of the ventricle, generating an upward-pointing dipole identifiable as the negative S wave. Then, the repolarizing effect of potassium currents in myocardial cells becomes counteracted by the opening of calcium channels, maintaining the depolarization in these cells for a prolonged duration. This depolarization plateau is visible on the ECG as the electrical silence during the ST segment. At the end of this segment, the calcium channels close, and the outward potassium currents repolarize the myocardial cells. This repolarization starts with the cells on the outer layer of the ventricular tissues. It propagates inward and generates a negative equivalent dipole pointing toward the negative lead. This last dipole is responsible for the T component of the ECG. This sequence of events is depicted in Figure 1.

Furthermore, we should note that atrial repolarization happens as the ventricle depolarizes. Since the ventricles contain many more cells than the atrium, it is responsible for a much larger equivalent dipole. This simultaneous activity obfuscates the effect of the atrium repolarization on the ECG.

The description of this process shows that ECG events offer a direct window into latent neural and ionic properties, as illustrated by the direct relationship between the width of the ST segments and the dynamics of the calcium channels generating the depolarization plateau in myocardial cells. We can exploit this relation when inferring from an ECG the properties of the cardiac process and its autonomic control.

## 3. Materials and Methods

### 3.1. Lognormal Modeling

For this work, we used an approach similar to [10] but adopting a sum of time-shifted and scaled lognormal density functions rather than scaled normal density functions. Conceptually, the lognormal density function is the same as the normal density function, with the time dimension being logarithmically transformed and, in our case, shifted by a time offset $t_{0}$. The lognormal density function is defined as in (Equation 1), where the index *j* can take the values {$P,Q,R,S,T_{+},T_{-}$} to refer to any of the components of the PQRST and the two lognormal components used for the T wave (i.e., $T_{+},T_{-}$). For convenience, we also defined $\Theta_{j}$ in (Equation 2) as the vector of parameters corresponding to the lognormal component $\Lambda_{j}$.
(1)$$\begin{array}{cc} \Lambda_{j}\left(t;\Theta_{j}\right) & =\frac{D_{j}exp\left(\frac{{\left[ln\left(t-t_{0j}\right)-\mu_{j}\right]}^{2}}{-2\sigma_{j}^{2}}\right)}{\sigma_{j}\left(t-t_{0j}\right)\sqrt{2\pi }} \end{array}$$
(2)$$\begin{array}{cc} \Theta_{j} & =\{\mu_{j},\sigma_{j},D_{j},t_{0j}\} \end{array}$$

### 3.2. ECG Sample

We collected 150 ECG recordings from 40 infants between 1 week and 24 months of age. A small device containing an ECG sensor (Actiwave Cardio; CamNTech) with two electrodes was placed on the infants’ chest. Raw ECG signals were recorded in a laboratory setting while the infants participated in different naturalistic experimental paradigms involving self-initiated interaction with objects or their parents.

Most recordings were sampled at 1024 Hz (N = 107). ECG signals recorded at a different rate (512 Hz: N = 13; 128 Hz: N = 30) were upsampled to 1024 Hz to ensure consistency. We segmented the data according to markers labeling the beginning and the end of a given experimental condition. Although we did not study the effect of experimental conditions for this paper, segmenting signals according to these markers ensured that we only kept valid ECG sections. For example, we automatically discarded initial data points recorded before the start of the experiment and before the sensor was placed on the infant’s chest. Most of the 555 segments obtained are of 5–25 min duration, but their distribution has a long tail, with the longest segment being 150 min long (percentiles (min): 0: 0.25; 10: 4.74; 25: 8.35; 50: 10.03: 75: 16.23; 90: 25.12; 100: 150.25).

### 3.3. Preprocessing

Heartbeats were automatically detected using HeartPy [12]. We rejected five segments because HeartPy could not process the signal (i.e., it made an exception) or because it detected less than 20 beats. For the remaining segments, we made beats comparable by epoching and normalizing the beat duration as follows. Considering $t_{1}$, $t_{2}$, and $t_{3}$, three subsequent R peaks, the epoched and normalized version of the peak corresponding to $t_{2}$ is obtained by (1) linearly interpolating the ECG between $t_{1}$ and $t_{2}$ over 250 regularly spaced samples, (2) doing the same for the segment from $t_{2}$ to $t_{3}$, and (3) concatenating these two segments. After this process, each beat is defined as one epoch containing a normalized ECG time series with 500 points starting at the previous R-peak and ending at the next R-peak, with its own R-peak centered in that window. For plotting, we map these epochs to the [−1, 1] interval and refer to the variable along that dimension as the *normalized time*. For each segment, we computed a *mean beat* ($\bar{x}$) by averaging across these epochs. We characterized the stability of the PQRST profile within a segment by computing the following signal-to-noise ratio (SNR):(3)$$\begin{array}{cc} SNR & =10*log_{10}\left(\frac{\sum_{i=1}^{500}{\bar{x_{i}}}^{2}}{\sum_{i=1}^{500}{\left(x_{i}-\bar{x_{i}}\right)}^{2}}\right) \end{array}$$

This definition is classic and widely used in the literature. It expresses in dB the ratio of the square amplitude of a signal over the square amplitude of noise (or error). The SNR characterizes how well the model fits the data. It is worth noting that an SNR of 0 means a noise amplitude as large as the signal, whereas an SNR of 20 means a noise amplitude $10^{2}$ smaller than the signal. These SNR values are computed for every beat and characterize how much these beats are similar to the average beat. By averaging these values across a segment, we obtain a value $\bar{SNR}$ that characterizes the quality of the signal (i.e., a noisy PQRST will be less similar, resulting in lower SNR values). Since the goal of this study is not to develop preprocessing techniques to clean noisy ECG signals, we rejected all segments with an $\bar{SNR}<5$ dB (N = 116). Figure 2 illustrates how the signal quality varies with $\bar{SNR}$. Eliminating noisy PQRST complexes allows us to concentrate on the specific contribution of this work, i.e., the model-driven analysis of the ECG. This exclusion criterion left 434 valid segments for analysis.

### 3.4. The PQRST Lognormal Prototype

The inverse modeling (i.e., curve fitting) constitutes a significant difficulty when modeling time series with sums of lognormal functions. Various algorithms have been developed to extract parameters for such models [13,14,15,16]. These extractors perform reasonably well, but the difficulty of this regression problem leaves room for improvement. For our application, we want to benefit from the ability to map lognormal components between time series provided by the prototype-based extraction [14]. We can adopt this approach when modeling time series with a stereotypical shape, as is the case for the PQRST complex. With that method, we embed our *prior* knowledge of the problem and our hypotheses about the generation of the observed signals by defining a starting solution (i.e., a *prototype*) made by a series of lognormal components. Then, we obtain the final solutions by adjusting this prototype to every sample, fitting the lognormal parameters to account for individual differences.

However, one issue with the least-square regression of prototypes is that such optimal solutions, if not properly constrained, can drift away from the conceptual modeling of the generative process captured by the prototype. For example, in this case, a lognormal component initially set in the prototype to model the P components can drift and end up fitting slight noise-related deviations in a high-amplitude component such as the R component. Although this is not a problem for obtaining a good curve fitting, it becomes an issue when studying the distributions of the parameters, e.g., associated with the P component. That is, although potentially providing an optimal curve fitting, such optimization may not hit the right balance between fitting the observation and providing a plausible solution for the biological or physiological mechanisms of the modeled process.

To help constrain our prototype, we used upper and lower bounds for the PQRST complexes. We defined these bounds analytically using box constraints on the lognormal parameters (Equation 4)–(Equation 7) (see [13] for the derivation of these bounds; notation further explained below). Figure 3 illustrates the envelope of all PQRST complexes generated by this *sigma-lognormal* (i.e., sum of lognormals) model, given the bounds we selected (see Table 1). We chose these bounds as a trade-off between encompassing the mean PQRST complexes of all recordings (except some apparent outliers) while constraining the possible solutions to minimize the proportion of the parameter space that can produce PQRST time series that are not physiologically plausible. We derived these bounds from the parameters $\Theta_{\Psi }$ (we use the index Ψ to refer to the prototype) of the prototype $\Sigma \Lambda_{\Psi }$ such that $\Theta_{\pm }=\Theta_{\Psi }\pm 0.2\left|\Theta_{\Psi }\right|$, except for (1) the value of the $D_{\pm }$ parameters being ceiled or floored to 0 when their interval intersects at 0, and (2) for $D_{T+}=D_{T_{+}\Psi }+0.4D_{T_{+}\Psi }$. The first exception ensures that the qualitative interpretation of the different waves remains intact. A sign inversion for a *D* parameter may mean changing a depolarization wave into a repolarization wave, or vice versa. Such a conversion is not possible in normal physiological conditions. For example, the P wave can only be due to the depolarization of the myocardial cells of the atrium. We will never observe a hyperpolarizing wave, rather than depolarizing wave, caused by an inversion of this process. Alternatively, such polarity reversal could be due to technical issues, such as a misplacement of the electrodes. However, these acquisition problems should be managed when preprocessing the recordings rather than being captured by the model parameters. We used the second exception for a pragmatic reason; we needed to allow the $D_{T}$ parameter to vary within a wider range to include most of the recorded PQRST complexes within the established bounds. Concerning the $|\Theta_{\Psi }|$ term, the |…| notation denotes the absolute value of $\Theta_{\Psi }$, and the indices in $\Theta_{\Psi }$, $\Theta_{+}$, and $D_{T_{+}\Psi }$ stand for “prototype”, “upper bound”, and “$T_{+}$ component of the prototype”, respectively. Other indices are derived using the same logic. The prototype was obtained by manually fitting a sigma-lognormal curve that captures the general trend of our sample of beat profiles. Its parameters are given in Table 2.
(4)$$\Lambda_{-}=\min\left\{\begin{array}{c} \Lambda (t-t_{0+};\mu_{+},\sigma_{-})\\ \Lambda (t-t_{0+};\mu_{+},\sigma_{+})\\ \Lambda (t-t_{0-};\mu_{+},\sigma_{+})\\ \Lambda (t-t_{0-};\mu_{-},\sigma_{+})\\ \Lambda (t-t_{0-};\mu_{-},\sigma_{-}) \end{array}\right.$$
(5)$$\Lambda_{+}=\left\{\begin{array}{ccc} 0 & if & t\leq t_{0-}]\\ \Lambda (t-t_{0-};\mu_{-},\sigma_{+}) & if & t\in ]t_{0-},t_{0-}+e^{\mu_{-}-\sigma_{+}}]\\ \Lambda (t-t_{0-};\mu_{-},\sigma =\mu_{-}-ln\left(t-t_{0-1}\right)) & if & t\in ]t_{0-}+e^{\mu_{-}-\sigma_{+}},t_{0-}+e^{\mu_{-}-\sigma_{-}}]\\ \Lambda (t-t_{0-};\mu_{-},\sigma_{-}) & if & t\in ]t_{0-}+e^{\mu_{-}-\sigma_{-}},t_{0-}+e^{\mu_{-}-\sigma_{-}^{2}}]\\ \Lambda (e^{\mu_{-}-\sigma_{-}^{2}};\mu_{-},\sigma_{-}) & if & t\in ]t_{0-}+e^{\mu_{-}-\sigma_{-}^{2}},t_{0+}+e^{\mu_{-}-\sigma_{-}^{2}}]\\ \Lambda (t-t_{0+};\mu_{-},\sigma_{-}) & if & t\in ]t_{0+}+e^{\mu_{-}-\sigma_{-}^{2}},t_{0+}+e^{\mu_{-}}]\\ \Lambda (t-t_{0+};\mu =ln\left(t-t_{0+}\right),\sigma_{-}) & if & t\in ]t_{0+}+e^{\mu_{-}},t_{0+}+e^{\mu_{+}}]\\ \Lambda (t-t_{0+};\mu_{+},\sigma_{-}) & if & t\in ]t_{0+}+e^{\mu_{+}},t_{0+}+e^{\mu_{+}+\sigma_{-}}]\\ \Lambda (t-t_{0+};\mu_{+},\sigma =ln\left(t-t_{0+}\right)-\mu_{+}) & if & t\in ]t_{0+}+e^{\mu_{+}+\sigma_{-}},t_{0+}+e^{\mu_{+}+\sigma_{+}}]\\ \Lambda (t-t_{0+};\mu_{+},\sigma_{+}) & if & t>t_{0+}+e^{\mu_{+}+\sigma_{+}} \end{array}\right.$$
(6)$$\Sigma \Lambda_{-}=\sum_{i}D_{i-}\left\{\begin{array}{ccc} \Lambda_{i-} & if & D_{i-}>0\\ \Lambda_{i+} & else \end{array}\right.$$
(7)$$\Sigma \Lambda_{+}=\sum_{i}D_{i+}\left\{\begin{array}{ccc} \Lambda_{i+} & if & D_{i+}>0\\ \Lambda_{i-} & else \end{array}\right.$$

### 3.5. Reinforcement Learning for Parameter Extraction

For this project, we used the Stable-Baselines3 and OpenAI Gym Python packages to implement a deep reinforcement learning approach for fitting our lognormal model to PQRST time series. We aim to improve over the grid optimization algorithm previously used to adjust the sigma–lognormal prototype to record time series. We adopted reinforcement learning for this problem because it can find optimal solutions to problems for which the “true” or “best” solution is not known, provided that we can define a measure of the quality of a solution. In our case, we can use the root-mean-square fitting error for that purpose. Contrary to classical supervised learning, reinforcement learning is not limited by the quality of the training data (i.e., it can adapt and become better than experts at a given task). Furthermore, the reinforcement learning approach is interesting due to the active development of Bayesian reinforcement learning [17]. We expect future Bayesian extensions to provide a more appropriate way to integrate prior knowledge into the estimation process by using prior distributions in the definition of the prototype instead of point values and box constraints (see the Discussion section).

Figure 4 shows a classic schematic representation of reinforcement learning, illustrating the interplay between the learning agent and its environment. In that paradigm, the policy maps the current state of the environment to the probability of performing given actions. This policy captures the transition probability of a Markovian decision process, as it does not depend on past states. In this context, the learning task boils down to optimizing the policy so that actions leading to the highest rewards have the highest probability of being executed. To implement such an approach, the space of possible actions, the observations space (i.e., the observable environment states), and the reward must first be defined.

**Action space:** Our model has 24 parameters. We consider these as latent variables. We defined the parameter space as a bounded box in $R^{24}$, using the upper bound $\Sigma \Lambda_{+}$ and the lower bound $\Sigma \Lambda_{-}$ previously defined. Furthermore, we consider the middle of this bounded box as a reference point, i.e., $\Sigma \Lambda_{ref}=\left(\Sigma \Lambda_{+}+\Sigma \Lambda_{+}\right)/2$. At the start of the fitting process, the estimated solution is set equal to $\Sigma {\hat{\Lambda }}_{0}=\Sigma \Lambda_{ref}$, where the index in $\Sigma {\hat{\Lambda }}_{n}$ indicates the step number, with 0 standing for the initial estimate. The action space is a box $A_{24}=\left\{\left(a_{1},a_{2},\ldots ,a_{24}\right)|a_{1},a_{2},\ldots ,a_{24}\in \left[-0.01,0.01\right]\right\}$. At each step of the fitting process, the action $a\in A_{24}$ taken by the agent is used to update the estimated solution following the rule $\Sigma {\hat{\Lambda }}_{i+1}=\Sigma {\hat{\Lambda }}_{i}+a\Sigma \Lambda_{ref}$. However, we note that even within this relatively narrow box, the sigma–lognormal parameters can take values such that the order of the PQRST lognormal components is changed (i.e., the Q component may move after the R component). Since such an inversion is contrary to the intent of the model, we further constrain this ordering. To this end, we first define the order of lognormal components by the timing of their peak. For a lognormal defined by parameters $\Theta_{j}$ (as defined in (Equation 2)), we can show this peak to happen at time $t_{pj}=t_{0j}+e^{\mu_{j}}e^{-\sigma_{j}^{2}}$. Thus, we constrain the impact of the action on $\Theta_{j}$ so that it is canceled whenever it results in $t_{pj+1}<t_{pj+1}$ (i.e., the action is such that it may cause the timing of the peaks of two consecutive components to become inverted), with *j* and j+1 referring to two consecutive lognormal components in the set of $\left\{P,Q,R,S,T_{+},T_{-}\right\}$ components. Alternatively, such restrictions could have been introduced within the reward function, e.g., by adding a penalty term that reduces the reward when these constraints are violated. However, constraining the problem that way would not preclude the algorithm from reaching solutions that violate these constraints. Since these solutions are biologically irrelevant, we preferred to limit the actions to categorically prevent them. We further limit the action so that the resulting parameters remain within the box defined by $\Sigma \Lambda_{+}$ and $\Sigma \Lambda_{-}$.**Observation space:** The observation space for this model is a dictionary containing values for the estimated parameters and the fitting errors. The fitting error space is a box of dimension 250. The PQRST signals are normalized to 500 points equally spaced between −1 and +1 (see the Preprocessing Section), but we observe only half of this interval, between −0.3 and 0.7. ECG signals in the remaining portion of the normalized time interval are mostly flat and contain little information. We set lower and upper bounds for that box at 1.5 times the minimum and maximum values observed for each of these points across all recorded PQRST time series. The observed values for the fitting error are taken as the difference between the PQRST time series being analyzed and the signal synthesized from $\Sigma {\hat{\Lambda }}_{i}$. The observation space for the estimated parameters is the 24-dimensional box bounded by $\Sigma \Lambda_{-}$ and $\Sigma \Lambda_{+}$, and the observed values are $\Sigma {\hat{\Lambda }}_{i}$. We further normalized the observation space, as is generally recommended when using a heterogeneous observation space. We confirmed that the normalization of the observation space resulted in a significant improvement in performance for our application.**Reward:** In reinforcement learning, the reward is the objective function to maximize and depends on the action performed. Thus, reinforcement learning algorithms aim at finding the action that maximizes the reward associated with any given state. For our application, we use as reward the difference in fitting SNR (as defined in (Equation 3)) before and after the action was taken.**Training:** The optimization of a PQRST time series terminates when the algorithm reaches its maximal number of steps (1000) or when the agent fails to find a solution (i.e., a set of parameter values) improving over its best SNR for 100 consecutive steps. Every time the optimization of a PQRST time series terminated, another time series was picked at random, and a new optimization was started. We trained our model for 3,000,000 steps, at which point performances were stable, and no additional training seemed likely to provide any advantage.

For learning, we implemented a deep reinforcement learning solution based on the Proximal Policy Optimization (PPO) [18] scheme. This relatively new algorithm has been designed to control the size of the steps taken when performing the gradient-based optimization of the policy. Large steps in the optimization process can result in instability as they can push the optimizer into a distant subspace with a flat objective function, potentially trapping the optimizer in an unproductive and low-reward region of the solution space. Specifically, in this approach, the update rule for the policy is defined by the loss function
(8)$$L\left(\theta \right)=E\left[\frac{\pi_{\theta }\left(a_{t}|s_{t}\right)}{\pi_{\theta_{old}}\left(a_{t}|s_{t}\right)}{\hat{A}}_{t}\right]=E\left[r_{t}\left(\theta \right){\hat{A}}_{t}\right]$$
where π represents the probabilistic policy parameterized with a new (θ) and an old ($\theta_{old}$) set of parameters, $a_{t}$ is the action selected at time step *t*, $s_{t}$ is the state of the environment at time *t*, and ${\hat{A}}_{t}$ is the advantage associated with a state, defined as the expected total reward discounted using a factor γ. This variable captures how much choosing the action $a_{t}$ when in state $s_{t}$ is better than randomly taking an action according to the current policy $\pi_{\theta }$, in terms of discounted reward (i.e., the future discounted reward when using a discounting factor γ is given by $r_{t}+\gamma r_{t+1}+\gamma^{2}r_{t+2}+\ldots$). We can recurrently formulate the value of a state in terms of expected future discounted reward using the Bellman equations [19]:(9)$$V\left(s_{t}|\pi_{\theta }\right)=\underset{\begin{array}{c} a_{t}∼\pi_{\theta },\\ s_{t+1}∼P \end{array}}{E}\left[r\left(s_{t},a_{t}\right)+\gamma V\left(s_{t+1}\right)\right]$$

We can similarly define an action-value function which is the same as $V(s_{t}|\pi_{\theta })$, but for a known action:(10)$$Q\left(s_{t},a_{t}|\pi_{\theta }\right)=\underset{s_{t+1}∼P}{E}\left[r\left(s_{t},a_{t}\right)+\gamma \underset{a_{t+1}∼\pi_{\theta }}{E}\left[Q(s_{t+1},a_{t+1})\right]\right]$$

Then, we can describe the advantage as a subtraction of these two functions
(11)$${\hat{A}}_{t}=Q\left(s_{t},a_{t}|\pi_{\theta }\right)-R\left(s_{t}|\pi_{\theta }\right)$$

Since the policy update specified by the rule (Equation 8) can be large (e.g., for $\pi_{\theta_{old}}\left(a_{t}|s_{t}\right)\approx 0$) and results in unstable learning, the PPO algorithm clip this function
(12)$$L^{C}\left(\theta \right)=E\left[min\left(r_{t}\left(\theta \right){\hat{A}}_{t},clip\left(r_{t}\left(\theta \right),1-\epsilon ,1+\epsilon \right){\hat{A}}_{t})\right)\right]$$
with epsilon set to a small value (e.g., around 0.2).

**Hyperparameter tuning:** We attempted to improve the learning and convergence of the PPO algorithm by tuning the hyperparameters using the Optuna-based [20] stable-baselines3-zoo package. The first attempt with 500 trials of 100,000 iterations failed to improve upon the default hyperparameterization provided with stable-baselines3’s implementation of PPO. We tried a second round with 1000 trials of 3,000,000 iterations with similar results. Consequently, we used the default hyperparameterization for PPO that comes with stable-baselines3.**Deep reinforcement learning and network architecture:** The PPO algorithm is implemented using deep learning and the architecture illustrated in Figure 5.

### 3.6. Parameter Denormalization

Since we estimated lognormal parameters in “normalized time”, we transformed them back into normal time before statistical analysis. For a lognormal profile to remain numerically identical after the compression of its time variable *t* by a factor α such that we have a new time variable $t^{*}=\alpha t$, we must adjust the original parameters Θ such that the new values $\Theta^{*}$ are defined as follows: $\left\{\mu^{*},\sigma^{*},t_{0}^{*},D^{*}\right\}=\left\{\mu +log\left(\alpha \right),\sigma ,\alpha t_{0},\alpha D\right\}$. Therefore, from the parameters Θ estimated using normalized time *t*, we computed the denormalized parameters using $\alpha =(t_{3}-t_{1})/2$, where $t_{1}$ and $t_{3}$ are the time of the R peaks for the preceding and following heartbeats, as previously defined (see Preprocessing Section).

### 3.7. Software

We used Python for all analyses. Preprocessing was performed mainly using HeartPy and MNE-Python and the standard Python scientific programming stack (NumPy, SciPy, Matplotlib, Seaborn, Pandas). The code used for the analyses is available at https://github.com/lina-usc/ecg_paper (accessed on 30 April 2023).

## 4. Results

### 4.1. Parameter Extraction

Once our deep neural network was trained using reinforcement learning, we used this system to extract lognormal parameters for all PQRST complexes from all segments (N = 751,510). For the final extraction, we increased the maximum number of iterations to 2000 and the maximum number of iterations with no progress to 200 to maximize our chances of obtaining solutions with high SNR. We excluded from further analyses beats fitted with an SNR<5 dB (18.28%). Figure 6 shows the distribution of fitting SNR values obtained with the proposed approach.

### 4.2. Use Case: Analysis of the Impact of Age

As a proof of concept for this approach, we investigated whether the modeling parameters are sensitive to a factor expected to impact ECG systematically: age. For this analysis, we average all beats in turns within segments and then within recordings. We rejected time points with recordings from less than eight participants to ensure a sufficient sample size for reliable statistical estimations. We used the remaining mean parameter values for statistical analysis (N = 121; per age (months): 1:9, 2:13, 3:19, 4:21, 6:14, 9:19, 12:17, 15:9).

Figure 7 illustrates the relationship between every modeling parameter and the age of the infant participants. Almost half of the parameters (10/24) show a statistically significant relationship with age with a *p*-value lower than 0.01 after correction for multiple tests. Furthermore, from the plot, some of these relationships (e.g., $t_{0}$ for the *S* component or μ for the $T_{-}$ component) are very clear, with very low *p*-values and large correlation coefficients.

Using a mixed-effect model with age as a fixed effect and the subjects as a random effect, we predicted the modification of the PQRST complex as a function of age. Figure 8 compares the prediction and the PQRST time series averaged across subjects using the mean and the median. As demonstrated by this figure, the effect of age is easier to summarize and understand using the model prediction than from the direct averaging of time series. This improved interpretability may reflect the “cleaning” of the data by imposing a linear variation of the parameters with age, effectively smoothing out natural intra- and inter-subject variability that may not be related to age. It should be kept in mind, however, that such a linear relationship is appropriate should be further validated. From some of the plots in Figure 7, an exponential relationship may better capture the variation of some of these parameters. This observation is particularly true of the $t_{0}$ parameters, which is unsurprising since time parameters such as reaction times tend to be distributed logarithmically. We can make a similar argument for amplitude parameters (i.e., *D*).

## 5. Discussion

Previously, we explained how the PQRST complex generated by the beating of the heart can be modeled by a set of superimposed lognormal components. Furthermore, we developed a novel way to perform parameter inference for this model using deep reinforcement learning. Then, we demonstrated that approach using the effect of age on an ECG as a use case. This analysis has shown that many modeling parameters have a systematic linear association with age. Notably, none of the parameters of the Q component exhibited statistical significance. Why no parameters of that specific components reached statistical significance is unclear. This may be due to a detrimental impact of the temporal superposition of the large neighboring R component on the estimation of the parameters of the component Q. Regardless, since we cannot conclude from a non-significant statistical test and given that this analysis is the first to look at the effect of age on this model, it would be premature to conclude on the lack of influence of age on the Q wave.

For this work, we chose to model the PQRST complex with lognormal rather than normal components. Multiple reasons motivate this choice. First, lognormal transfer functions are causal. They have a support that starts at a specific time $t_{0}$, as opposed to an ]−∞,+∞[ support for the normal equation. Furthermore, the use of sums of lognormal functions for modeling biological time series has been well-developed for the study of motor control [21,22,23,24]. The biological relevance of the lognormal as a transfer function is well-rooted mathematically through the central limit theorem, as applied to biological networks of subprocesses coupled through proportionality. In contrast, the response of a system made of independent and additive subprocesses converges, under the same theorem, toward a normal response. These ideas have been developed in detail in the context of the lognormality principle [11,25].

In terms of interpreting the meaning of lognormal parameters, $t_{0}$ (in seconds) represents the time of initiation of the lognormal process. Since lognormal-shaped waveforms are emerging from the convergence of small effects associated with many coupled subsystems, we can see $t_{0}$ as the moment the first cells generated an action potential and triggered an avalanche of subsequent action potentials in the network of neighboring cells. The parameter *D* represents the amplitude of the equivalent electric dipole created by this process. More precisely, it is equal to the integration over time of the depolarization or repolarization waves (in V∗s). The parameters μ and σ represent, on a logarithmic scale, the time delay (i.e., how long it takes to respond) and response time (i.e., how spread the response is in time) of the network of excitable cells. Therefore, these parameters are emergent properties due to the speed of propagation of the electrical waves in these networks. We expect their values to reflect biological properties that modulate the propagation speed, such as the strength of gap-junction coupling between neighboring cells.

Six partly overlapping lognormal equations were used to model the PQRST complex. The P, Q, R, and S waves are all represented by a single equation. For expediency, we accounted for the negative skew of the T wave by representing it with the subtraction of two lognormals (noted $T_{+}$ and $T_{-}$). Further investigation will be required to understand the physiological underpinning of this negative skew and, consequently, the most biologically relevant way to model it.

With respect to fitting accuracy, SNR values obtained with our model are lower than those reported for a similar modeling approach used for analyzing human movements. For example, an average SNR of 20.75 dB was reported for a prototype-based lognormal modeling of the speed of triangular motion [14]. We believe that this lower fitting accuracy for ECG signals is partly due to systematic offsets in the resting potentials (e.g., see the segments before the QRS complex in Figure 6b). Such systematic offsets significantly contribute to the modeling error and can be observed at steady state for electric potential but not for the speed of human movements. We expect the fitting accuracy from an approach such as [10] to be higher than what was obtained with our model, although we did not explicitly compare accuracies. Published values may not be comparable because they were obtained on a different dataset, with different preprocessing, targeting different populations. Furthermore, the two approaches are not comparable. For example, ours uses only 24 parameters, whereas [10] uses 35. This approximative 50% increase in the number of modeling parameters is expected to provide more flexibility to improve fitting accuracy. More importantly, we aimed to develop a biologically relevant model rather than obtain maximal fitting accuracy. High fitting accuracy is highly desirable for some applications (such as the signal compression application mentioned in [10]). However, for physiological interpretability, the biological relevance of the model and the preservation of component order are more important and should be prioritized even when it results in some loss in fitting accuracy. These arguments should be familiar to anyone familiar with the issue of model overfitting.

Furthermore, concerning training and validation, it should be noted that we did not use a hold-out method or cross-validation for this application since our ECG model was fixed and we wanted to train an estimator that would provide the best parameter fitting. Our goal was not to release a generalizing extractor that would perform well on a new dataset, to benchmark classification metrics (e.g., accuracy), or any similar application that would require such controls. Such a lack of cross-validations is not infrequent in the context of classical reinforcement learning (e.g., obtaining the best performance at a game) when a generalization is not required.

Our overarching aim in this paper was to demonstrate how the analysis of an ECG can benefit from using a model-driven approach. Most biosignals relevant to fundamental science and practical applications in medicine and neuroscience present us with ill-posed problems due to the complexity of their generative processes, the large quantity of uncontrolled latent variables, and the relative sparsity of experimental measurements. To progress in the study of such systems, we need to constrain our analysis using prior knowledge. We can best exploit such knowledge by embedding it in the structure and parameterization of our models. This process effectively injects our knowledge into our analysis by operationalizing our hypotheses about the generative mechanisms that caused the observed signals. We can deploy similar approaches for a wide range of problems. For example, dynamic causal modeling (DCM) constitutes a similar model-driven approach for the study of the brain [26]. Concerning more specifically the model proposed here, we believe that such approaches may not only be useful for fundamental research on the generation of an ECG but can also find clinical applications for biomarker development. In this case, modeling parameters can be used directly as features for classifying particular diseases. For example, the absence of a Q wave has been considered to suggest abnormal left ventricular diastolic function [27]. Such a condition would therefore result in the $D_{Q}$ parameters’ value being negligible. Similarly, many other conditions have specific effects on the different segments or waves constituting the PQRST complex. Therefore, these conditions are likely to have a systematic effect on the parameters of our model. Researchers interested in using our approach for such applications should follow the approach we illustrated (e.g., see Figure 3) to adjust the constraints on the model (Table 1) to include the type of deviations from the typical profile of the PQRST complex they want to study.

We plan to develop this framework further along three lines of investigation. First, here we demonstrated this approach using a partially mechanistic model (e.g., lognormals are associated with known electrical processes in the heart) and partially phenomenological (e.g., we used the subtraction of two lognormals to accommodate the reverse asymmetry of the T wave). The full power of such modeling can be reached only when using rich mechanistic models. Phenomenological models can be useful in practical applications. For example, they may offer ways to summarize the data that help improve performances for biomarkers in clinical applications. However, we argue that, for fundamental research, a model is as good as its relationship to the underlying mechanisms. By developing models that mechanistically map latent variables to observable signals, we gain ways to make inferences on previously inaccessible variables. Refining such models by properly capturing the various biological mechanisms is an incremental and protracted process. In our case, we have good reasons to assume the lognormality of the different waves of the PQRST complex [11,25]. Nevertheless, further analyzing how the physiology and anatomy of the heart impact the shape and lognormality of these waves may improve the interpretability and conceptual validation of this approach. This investigation will, for example, require a more in-depth analysis of the propagation of polarization/depolarization waves along the atrial and ventricular myocardial cells, and the potential impact of border effects as these waves travel through these spatially bounded structures.

Second, for this novel approach we used simple box constraints on the parameter space for our model as is usual for this kind of application. Such binary (as opposed to probabilistic) constraints are relatively unrealistic and unrefined. Box constraints enforce the idea that the probability of the parameters taking a given value goes from being null to uniformly distributed for an infinitesimal difference in parameter values. Such a discontinuity is not a fair representation of our prior knowledge or assumptions. Furthermore, box constraints generally do not accurately capture the subspace of plausible or acceptable solutions. The importance of this problem increases with dimensionality. The proportion of space occupied by the “corners” of the box (as opposed to its center) increases with the number of dimensions. These corners represent portions of the subspace where multiple parameters have extreme values (i.e., edge cases). Corresponding solutions are, therefore, likely to be biologically implausible. This effect is not negligible in 24 dimensions. For example, whereas a unit-size ball occupies 52% of a unitary cube, a unit-size hyperball occupies 0.00000001% of a unit-size hypercube in a 24-dimensional space. Although such development was out of the scope of this particular work, for future work, we propose to integrate a Bayesian approach to constrain the estimation of parameters using prior distributions. Aside from being more realistic, such constraints on modeling parameters may allow for an easier incremental integration of new knowledge by adjusting the Bayesian priors. It also integrates more naturally into hierarchical models, where such priors can be generated as an output of another part of the model. Since we do not expect the manifold supporting plausible solutions to be box-shaped, such prior distributions may ideally be a multivariate distribution rather than a series of 24 univariate distributions.

Third, although we consider our approach to be highly relevant for many applications related to cardiology, we primarily aim to develop this model to support the use of the ECG as a window into the state of the autonomic nervous system. Thus, we plan to model the generation of the heartbeat sequence rather than independent PQRST complexes, as proposed in this paper. The heartbeat is regulated by the balance between the opposing sympathetic (epinephrine and norepinephrine) and parasympathetic (acetylcholine) systems. Both systems influence the cardiac process by modulating the opening and closing of calcium channels in myocardial cells. By modeling the impact of these inputs on the heart, the analysis of the ECG could provide insights into autonomic control and open the door to clinical applications in conditions wherein this control is atypical, such as autism spectrum disorder [28].

## 6. Conclusions

By embedding our knowledge into a forward model of ECG, we demonstrated how a sum of overlapping lognormal can be used to analyze the ECG and the properties of its PQRST complex. We also provided a framework to estimate the values of modeling parameters using deep reinforcement learning, and we constrained this optimization to preserve parameter interpretability (i.e., parameters are not allowed to take values that may change their qualitative interpretation). Finally, we used the effect of age to test the sensitivity of the modeling parameters to factors systematically affecting the ECG. In our discussion, we highlighted some directions to explore to benefit fully from the potential of this approach. We are confident that, as scientists and engineers come to refine their understanding of the complexity of the various natural systems they are tackling, the systematic embedding of our knowledge into mechanistic models and the adoption of a model-driven analytic approach will increasingly reveal itself to be the surest way forward. 

## Figures and Tables

**Figure 1 bioengineering-10-00696-f001:**
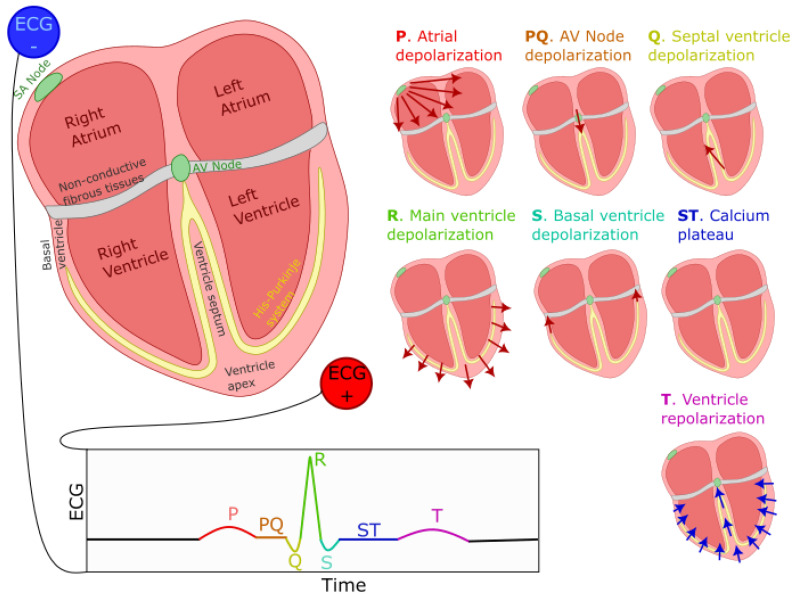
Generation of the ECG PQRST complex. The left–right image shows the positioning of ECG electrodes. Seven smaller representations of the heart illustrate the direction and polarity of electrical dipoles generated at each heartbeat phase. The bottom of the figure displays the resulting PQRST waveform. Red (blue) arrows indicate the direction of depolarization (repolarization) waves.

**Figure 2 bioengineering-10-00696-f002:**
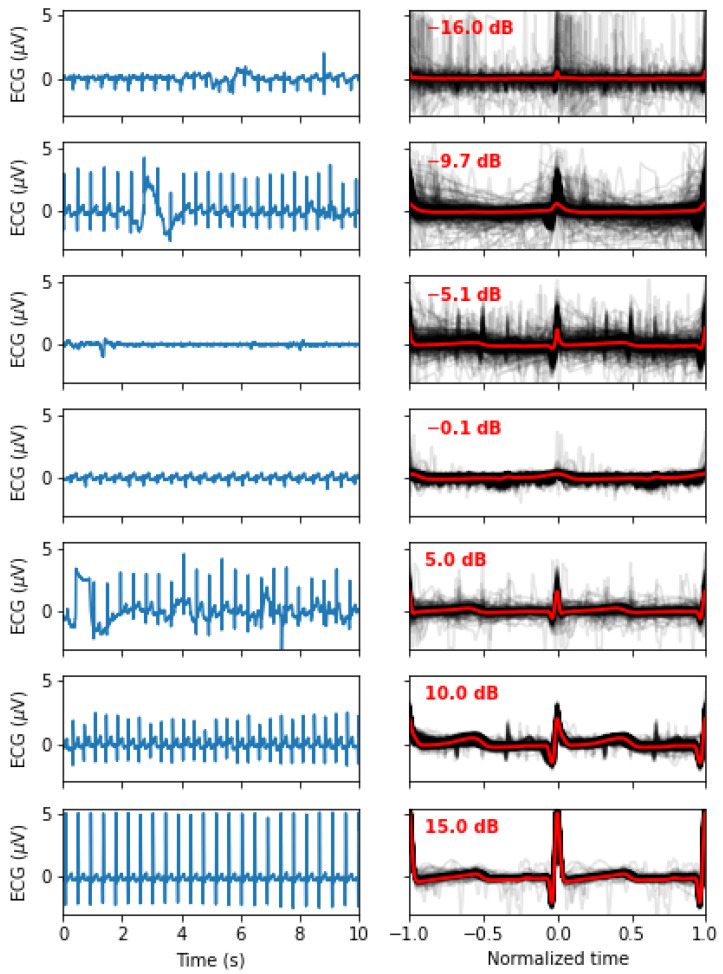
Signal quality as a function of the $\bar{SNR}$, as defined in (Equation 3). The left panels show 10 s of the original signals, whereas the right panels show all the normalized beats (in light grey) and the mean beat (red). From top to bottom, we display examples of recordings with an $\bar{SNR}$ closest to −15, −10, −5, 0, 5, 10, and 15 dB.

**Figure 3 bioengineering-10-00696-f003:**
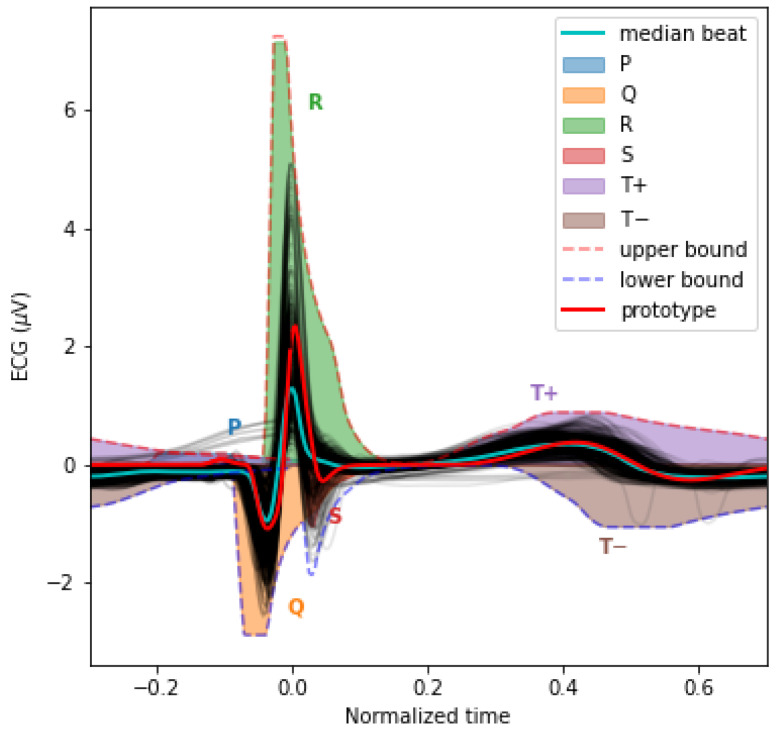
This plot overlays the mean PQRST for each valid segment (light grey) and their median (cyan). We also overlaid the sigma-lognormal prototype (red) and the upper (dashed red) and lower (dashed blue) envelop of the PQRST time series that can be generated by the model when using parameters within the $\Sigma \Lambda_{-}$ and $\Sigma \Lambda_{+}$ bounds specified in Table 1. We also show the space covered by each component in different colors.

**Figure 4 bioengineering-10-00696-f004:**
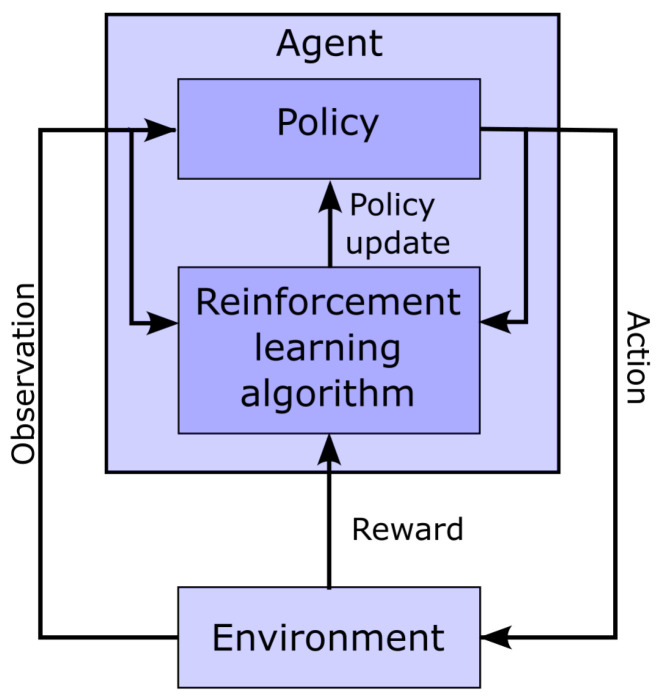
Schematic representation of the interaction between the agent and the environment in the context of reinforcement learning.

**Figure 5 bioengineering-10-00696-f005:**
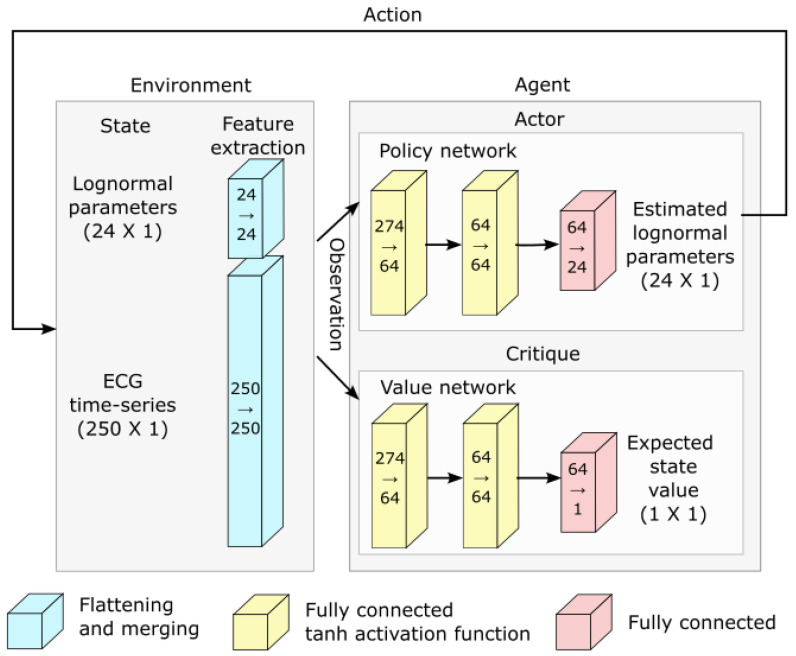
Network architecture used for deep reinforcement learning.

**Figure 6 bioengineering-10-00696-f006:**
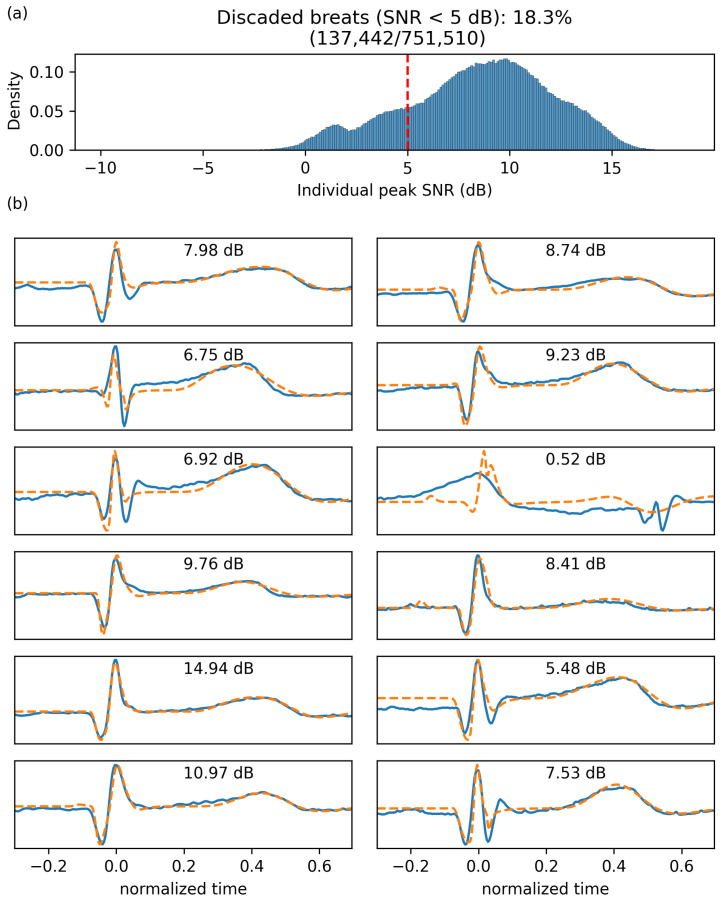
Fitting SNR. (**a**) Distribution of fitting SNR for individual beats. (**b**) Twelve randomly selected beats (solid blue) with their fitting (dashed orange) and the corresponding SNR.

**Figure 7 bioengineering-10-00696-f007:**
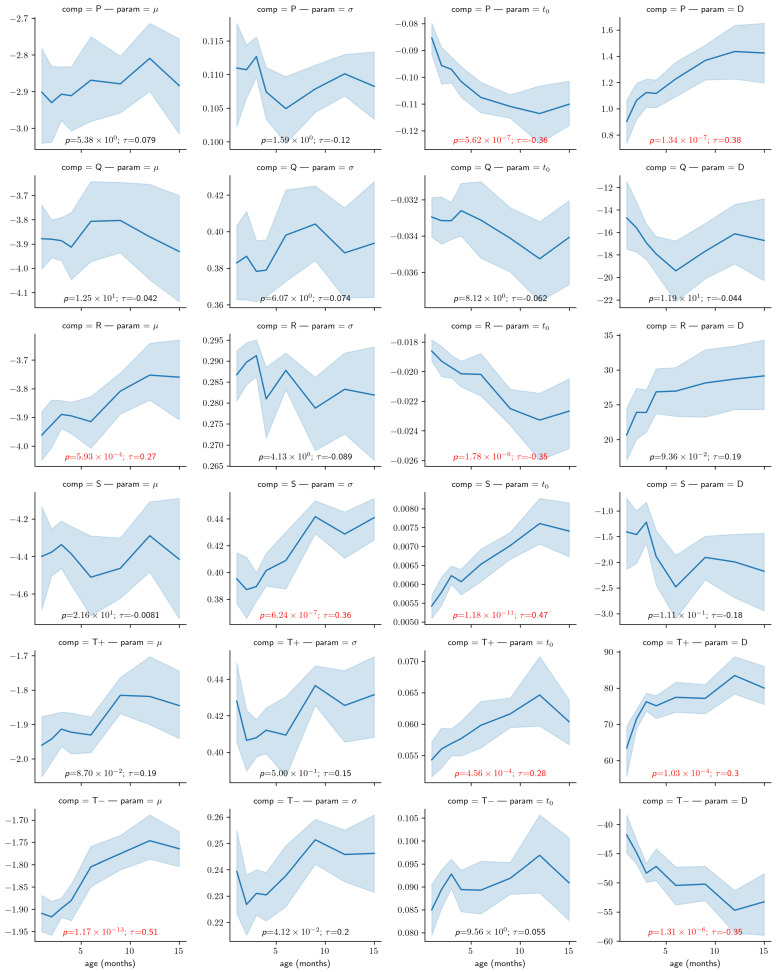
Correlation between the age and the modeling parameters. The light blue region denotes the 95% confidence interval obtained with bootstrapping. Overlaid on each panel is the Kendall rank correlation coefficient (τ) and its corresponding *p*-value corrected for 24 independent tests using Bonferroni’s correction (i.e., reported *p*-values have been multiplied by 24), both in red if significant (i.e., p<0.01). Parameters *D* are expressed in μV ∗ s (i.e., multiplied $1\times 10^{6}$), for convenience.

**Figure 8 bioengineering-10-00696-f008:**
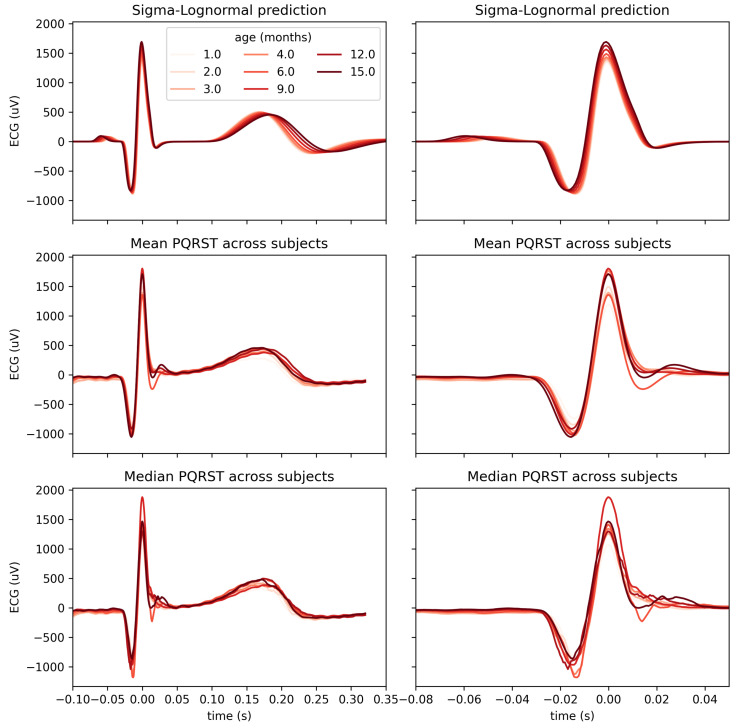
Comparison of PQRST complexes predicted from Sigma–Lognormal modeling (top panel), mean (middle panel), and median (bottom panel) PQRST across participants, for each time point with at least 8 participants. Both columns show the same data; the right columns only zoom further on the P peak to better appreciate the modulation of its amplitude by age.

**Table 1 bioengineering-10-00696-t001:** Parameter bounds. $D_{\pm }$ are in μV and $t_{0\pm }$ are using normalized time.

	$\mu_{-}$	$\mu_{+}$	$\sigma_{-}$	$\sigma_{+}$	$t_{0-}$	$t_{0+}$	$D_{-}$	$D_{+}$
*P*	−2.4	−1.6	0.08	0.12	−0.288	−0.192	0	3.6
*Q*	−3.6	−2.4	0.32	0.48	−0.096	−0.064	−60	0
*R*	−3.6	−2.4	0.2	0.3	−0.054	−0.036	0	96
*S*	−4.2	−2.8	0.32	0.48	0.012	0.018	−12	0
$T_{+}$	−1.2	−0.8	0.32	0.48	0.12	0.18	0	204
$T_{-}$	−1.2	−0.8	0.184	0.276	0.176	0.264	−144	0

**Table 2 bioengineering-10-00696-t002:** PQRST Sigma–Lognormal prototype. $D_{\pm }$ are in μV and $t_{0\pm }$ are using normalized time.

	μ	σ	$t_{0}$	*D*
*P*	−2.0	0.1	−0.24	3
*Q*	−3.0	0.4	−0.08	−50
*R*	−3.0	0.25	−0.045	80
*S*	−3.5	0.4	0.015	−10
$T_{+}$	−1.0	0.4	0.15	150
$T_{-}$	−1.0	0.23	0.22	−120

## Data Availability

Data are not shared at this time because the data collection is ongoing. The code used for the analyses is available at https://github.com/lina-usc/ecg_paper (accessed on 30 April 2023).

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
