# Peer review of "Model-Driven Analysis of ECG Using Reinforcement Learning"

_bioengineering, 2023, doi:10.3390/bioengineering10060696_

Round 1

Reviewer 1 Report

The authors proposed a novel methodology to model ECG based on the sum of lognormal function and used reinforcement learning for parameter optimization. Correlation could be seen between the age and the modeling parameters. However, there are a few concerns which needed extra explanation or revision in details.

1.         What is the exact difference between lognormal function and normal function? As far as I know, previous researches has used the set of normal function to fit ECGs, so what are the advantages if lognormal function was used?

2.         Authors computed the bounds of parameters before optimization according to formula (4)-(7). More explanation should be made for these formula since multiple indices were involved.

3.         A few words probably lead to misunderstandings, for example, “500-sample epochs” in line 211.

4.         The proposed method was reasonable theoretically. I would like to encourage the authors to discuss further implications of their methodology and how to use it in the clinical environment and practical engineering.

Reviewer 2 Report

Christian O’Reilly et al. presented a model-driven method of the analysis of ECG and demonstrated how a sum of overlapping lognormal can be used to analyze the ECG and the properties of its PQRST complex. They also provided a framework to estimate the values of modeling parameters using deep reinforcement learning. The manuscript is basically complete and the logic of the story is clear. I recommend minor revision before acceptance of this paper as exemplified below:

(1) The authors have not clearly explained what the function is between signal quality and the SNR (Figure 2).

(2) The authors have not clearly explained why the results are not significant when the “comp=Q” (Figure 6).

(3) In the Discussion, the authors stated that “Such binary (as opposed to probabilistic) constraints are relatively unrealistic and unrefined.” So why are box constraints used on the parameter space for the model?

(4) The format of the References is inconsistent.

Good.

Reviewer 3 Report

This paper presents a reinforcement-learning based method to fit the sum of lognormal to a single ECG heartbeat. The approach is interesting, and it has many potential applications, such as generating realistic ECG signals.   A big issue is that the paper did not report how well the sum of lognormal fits the data: There is no quantitative result to show the goodness/error of curve fitting, such as R2 or mean-squared-error   other comments:   typo in line 206: detexted  => detected   the 206-207: we epoched and normalized beat duration to make beats comparable. Please explain the detailed methods for 'epoched and normalized'.   line 332: the reward is defined as the difference in fitting SNR before and after the action Please give detailed description with equations, because reward function is the most importance component of reinforcement learning.   in Equation (9),  St+1 ~ P,  what is the meaning of 'P'? it is not explained.   "However, we note that even within this relatively narrow box, the sigma-lognormal parameters can take values such that the order of the PQRST lognormal components gets changed (i.e., the Q component may move after the R component)." Could  this be avoided by modifying the reward function?  

Round 2

Reviewer 1 Report

I think the revised manuscript has improved. I have no further comments.

Reviewer 3 Report

NA

NA